# Alterations in Progesterone Receptor Isoform Balance in Normal and Neoplastic Breast Cells Modulates the Stem Cell Population

**DOI:** 10.3390/cells9092074

**Published:** 2020-09-11

**Authors:** María Sol Recouvreux, María Inés Diaz Bessone, Agustina Taruselli, Laura Todaro, María Amparo Lago Huvelle, Rocío G. Sampayo, Mina J. Bissell, Marina Simian

**Affiliations:** 1Área de Investigaciones, Instituto de Oncología, “Ángel H. Roffo”, Av. San Martín 5481, Buenos Aires C1417DTB, Argentina; mrecouvreux@mednet.ucla.edu (M.S.R.); ine.db84@gmail.com (M.I.D.B.); ma.taruselli@gmail.com (A.T.); ltodaro@gmail.com (L.T.); 2Department of Obstetrics and Gynecology, David Geffen School of Medicine, University of California Los Angeles, Los Angeles, CA 90095, USA; 3Instituto de Nanosistemas, Universidad Nacional de San Martín, 25 de Mayo 1021, San Martín, Buenos Aires 1650, Argentina; mariamparo86@gmail.com; 4Department of Bioengineering, University of California, Berkeley, CA 94720, USA; ro.sampayo@gmail.com; 5Division of Biological Systems and Engineering, Lawrence Berkeley National Laboratory, Berkeley, CA 94720, USA; mjbissell@lbl.gov

**Keywords:** progesterone receptor isoforms, mammary gland, breast cancer, estrogen receptor, stem cells

## Abstract

To investigate the role of PR isoforms on the homeostasis of stem cells in the normal and neoplastic mammary gland, we used PRA and PRB transgenic mice and the T47D human breast cancer cell line and its derivatives, T47D YA and YB (manipulated to express only PRA or PRB, respectively). Flow cytometry and mammosphere assays revealed that in murine breast, overexpression of PRB leads to an increase in luminal and basal progenitor/stem cells. Ovariectomy had a negative impact on the luminal compartment and induced an increase in mammosphere-forming capacity in cells derived from WT and PRA mice only. Treatment with ICI 182,780 augmented the mammosphere-forming capacity of cells isolated from WT and PRA mice, whilst those from PRB remained unaltered. T47D YB cells showed an increase in the CD44^+^/CD24^Low/−^ subpopulation; however, the number of tumorspheres did not vary relative to T47D and YA, even though they were larger, more irregular, and had increased clonogenic capacity. T47D and YA tumorspheres were modulated by estrogen/antiestrogens, whereas YB spheres remained unchanged in size and number. Our results show that alterations in PR isoform balance have an impact on normal and tumorigenic breast progenitor/stem cells and suggest a key role for the B isoform, with implications in response to antiestrogens.

## 1. Introduction

The normal mammary gland consists of an epithelial hierarchy where tissue-specific stem cells are able to self-renew and produce committed progenitors that give rise to terminally differentiated cells which comprise the functional gland [1]. The epithelium of the mammary gland is composed of two cellular lineages: luminal cells that surround a central lumen and myoepithelial cells that are located in a basal position next to the basement membrane. Together, these cells are organized into a series of branching ducts that end in secretory alveoli that produce milk during lactation. The mammary gland is highly sensitive to the effects of the ovarian hormones—estrogen and progesterone—that have a direct impact on postnatal development, estrous cycling, pregnancy, lactation, and involution [2]. Even though mammary stem/progenitor cells do not express steroid receptors, paracrine effectors impact stem/progenitor cell homeostasis [3].

Estrogen receptor-alpha (ERα) is critical for mammary gland development and plays a key role in tumor progression and in determining whether a patient is eligible for endocrine therapy; 75% of diagnosed breast cancers are ERα-positive [4,5]. Estrogens are proposed to play an indirect role in the expansion of normal murine mammary stem cells by inducing PR expression [6,7]. Several papers have shown that estrogens do not increase the pool of breast cancer stem cells directly [8,9]. However, selective ER down modulators, like Tamoxifen, down regulators such as ICI 182,780, and aromatase inhibitors have proved to enrich the cancer stem cell pool [10,11].

The progesterone receptor (PR), like ER, belongs to the superfamily of steroid receptors and mediates the action of progesterone in its target tissues [12]. In particular, progesterone, acting through its cognate receptors, has been shown to regulate the proliferation of mammary stem cells via paracrine signaling in mice [6,7]. Additionally, progesterone and synthetic progestins increase populations of breast cancer stem cells in human breast cancer cell lines [8,13,14]. PR exists in two isoforms, A and B, that are transcribed from a single gene, and the expression of these, in an appropriate ratio, is critical for normal mammary development [15]. Mammary glands of transgenic mice carrying additional PRA are characterized by extensive lateral branching, ductal hyperplasia, a disorganized basement membrane (BM), and loss of cell–cell adhesion [16,17]. On the other hand, mammary glands of mice carrying additional PRB are characterized by reduced branching morphogenesis, although alveologenesis is normal when pregnancy occurs [18]. The role of PR signaling in breast cancer development and progression is controversial. In multiple rodent models, deletion or inhibition of the PR pathway results in significant reduction in mammary carcinogenesis [19,20,21,22]. The two isoforms are generally co-expressed at similar levels in the normal breast, but the ratio can be altered in human breast tumors, resulting in a predominance of one isoform, usually PRA, over its counterpart [23,24,25]. More recent papers showed that PRB-expressing tumors are associated with more advanced disease [26,27]. Thus, understanding the relative contribution of each PR isoform to these observations is of critical importance to deepen our understanding of breast cancer etiology.

To further investigate the role of PR signaling in the homeostasis of the stem cell population of the mammary gland and breast cancer, we used PRA and PRB transgenic mice and the T47D human breast cancer cell line and the sublines derived from it that express either PRA (T47D YA) or PRB (T47D YB). Additionally, we looked into the role of estrogen signaling on the stem cell population in the context of PR isoform imbalance. Our results show that PRB is a stronger driver of the stem cell population than PRA and that ER signaling might be compromised in this context.

## 2. Materials and Methods

### 2.1. Mice, Treatments with Steroids and Tissue Preparations

The mice used in these studies were of the FVB strain. All animal studies were conducted in accordance with the standards of animal care as outlined in the NIH and ARRIVE Guidelines for the Care and Use of Laboratory Animals. All animal procedures were approved by the Comité Institucional para el Cuidado y Uso de Animales de Laboratorio (CICUAL) Instituto de Oncología “Angel H. Roffo” on 5 October 2012, protocol #2012/08. Animals were killed by carbon dioxide inhalation and cervical dislocation at the indicated times. PRA and PRB transgenic mice have been described previously [16,17,18,28]. Nulliparous adult (20 to 22 weeks old) mice were either sham operated or ovariectomized and/or treated with 50 μg/mouse of ICI 182,780 (Tocris Cookson Inc., Ellisville, MO, USA) daily for 4 days, 15 days after the surgical intervention. For mammary whole mounts, one of the inguinal mammary glands was fixed in Carnoy’s solution and stained in Alum Carmine (0.2% carmine/0.5% aluminum potassium sulfate (both from Sigma, St. Louis, MO, USA)).

### 2.2. Mammary Gland Primary Cultures

Mammary glands were minced using razor blades and digested in 10 mL digestion media (digestion buffer containing DMEM/F12, 100 mg/mL gentamycin and 0.2% collagenase type IV (Sigma-Aldrich), 0.2% trypsin and 5% FBS). Samples were incubated at 37 °C for 8 to 10 h, on a rotary shaker. Cells were washed with DMEM/F12 10% FBS twice or until the supernatant was clear in the presence of DNase (Sigma-Aldrich). Cells were washed with lysis buffer (0.64% NH_4_Cl) to eliminate blood cells. A final wash with DMEM/F12 without serum was performed in order to remove serum before the cells were used for mammosphere assays or flow cytometry.

### 2.3. Cell Culture

T47D cell lines (T47D, T47D YA, and T47D YB) were kindly provided by Dr. Carol Lange. They were routinely maintained in growth medium, consisting of DMEM/F12 (Sigma-Aldrich), supplemented with 10% fetal bovine serum (FBS, Internegocios, Buenos Aires, Argentina) and gentamicin, in a humidified 5% CO_2_/air atmosphere. Serial passages were carried out by treatment of 80% confluent monolayers with 0.25% trypsin (Invitrogen) and 0.02% EDTA in Ca_2_-free and Mg_2_-free PBS. T47D-YA and T47D-YB cells were additionally supplemented with 200 µg/mL G418. Hormone treatments were carried out with charcoal-stripped serum and phenol red-free DMEM/F12, as previously described [29].

### 2.4. Mammosphere Assays

Single cells derived from mammary glands from transgenic or T47D cell lines were plated in 12-well low attachment suspension culture plates (Greiner Bio-One, Koln, Germany) at a density of 10,000 or 1000 viable cells/mL, respectively. Cells were grown in 1 mL serum-free media, supplemented with B27 (Gemini Bioproducts, West Sacramento, CA, USA), and 20 ng/mL EGF, as previously described [10,30]. When indicated, mammosphere cultures were treated with 17-β-estradiol (Santa Cruz Biotechnology, Dallas, TX, USA) at a concentration of 10^−8^ M and ICI 182,780 (Santa Cruz Biotechnology, Dallas, TX, USA) at a concentration of 10^−6^ M. Both were prepared in absolute ethanol, which was used on its own as vehicle control. Mammospheres were counted after 7–10 days in culture with a Nikon eclipse TE2000-S inverted microscope. To calculate diameters and sphericity, mammosphere images were analyzed using Image J. Sphericity was calculated as longest diameter/shortest diameter; at least 15 mammospheres were measured per experiment and condition. Experiments were repeated at least 3 times.

### 2.5. Flow Cytometry

Single cell suspensions from mouse mammary glands or T47D cell lines were washed and resuspended in PBS containing 2% FBS at a concentration of 500,000 cells/mL. In the case of primary mouse mammary cells, a mouse lineage panel (PE anti-mouse Lineage Cocktail, BioLegend, San Diego, CA, USA) used in a 1/7 dilution, designed to react with cells from the major hematopoietic cell lineages, was used to select the Lin^−^ population. CD29-FITC (1/100) and CD24-APC (1/300) (both from Biolegend) were used to identify basal and luminal cell compartments. For human breast cancer cells, CD24-FITC (1/100) and CD44-APC (1/300) (Biolegend) were used. In all cases, cells were incubated for 1 h on ice and subjected to FACS analysis (FACS Area, FCEN-UBA), followed by Flow Jo analysis. The Lin^−^CD24^+^/CD29^hi^ fraction is enriched in mammary basal and mammary stem cells, whereas the Lin^−^CD24^+^/CD29^+^ population comprises luminal progenitor cells and mature luminal cells [31]. Human breast cancer stem cells are characterized by being CD44^+^/CD24^Low/−^ [32].

### 2.6. Clonogenic Assays

Mammospheres were enzymatically dissociated with 0.05% trypsin for 15 min at 37 °C to obtain a single-cell suspension and were then plated at low density (10,000 cells/mL) in 6-well plates. After 15 days, cells were stained with 1% crystal violet, and colonies containing more than 30 cells were counted.

### 2.7. Immunofluorescence

Mammospheres were fixed in PFA 4% for 20 min at RT, permeabilized with Triton X-100 1% in PBS for 10 min at 4 °C, then blocked in BSA 3%. Primary antibodies against ER (Santa Cruz Biotechnology, Dallas, TX, USA) were incubated at a 1:100 dilution overnight at 4 °C. The secondary antibody, anti-rabbit Alexa 488 (Abcam), was incubated at RT for 1 h. Propidium iodide was used for nuclear staining. Nonspecific binding was address by incubating with secondary antibody alone. Images were taken by confocal microscopy (Olympus FV-1000) and analyzed with Image J.

### 2.8. Western Blots

Protein extracts were prepared as previously published by our laboratory; cells were homogenized on ice in RIPA buffer (50 mM Tris, pH 8.0 containing 150 mM NaCl, 0.1% SDS, 0.5% deoxycholate, and 1% NP40) containing protease inhibitors. Protein concentrations were measured using the Bradford method. Samples were mixed with loading buffer containing β-mercaptoethanol and boiled for 5 min. One-hundred micrograms of each sample were then separated in SDS-PAGE mini gels (BioRad) and transferred to PVDF membranes (Amersham Biosciences, Uppsala, Sweden). The membranes were incubated for 1h at room temperature in blocking buffer (5% fat free milk, 0.1% Tween-20 in PBS). Primary antibodies were used at a 1/200–1/2000 dilution in blocking buffer and were incubated at 4 °C overnight. After washing with PBST, membranes were incubated with secondary antibodies at a 1/1000 dilution for 1 h at room temperature. Signals were detected with an enhanced chemiluminescence kit (ECL, Amersham Biosciences). Primary antibodies: rabbit anti-ER alpha, rabbit anti-E-cadherin, rabbit PR (Santa Cruz Biotechnology, Dallas, TX, USA). Secondary antibody: donkey-anti-rabbit HRP.

### 2.9. Statistical Analysis

Significant differences among assays were identified by the Student’s *t*-test, one or two-way ANOVA, followed by Bonferroni’s comparisons test. A value of *p* < 0.05 was considered significant.

## 3. Results

### 3.1. PRB Overexpression Alters the Luminal and Basal Cell Compartments in the Normal Mouse Mammary Gland

To study the impact of PR isoform balance in the mammary gland stem cell population, we used the previously characterized transgenic mice overexpressing either the A or B isoforms of PR [16,17,18,28]. The change in the balance of the PRA/PRB ratio leads to aberrant outcomes; while overexpression of PRA promotes hyperplasia in the mammary gland, PRB overexpression reduces branching morphogenesis in about 20% of cases compared to mammary glands of WT mice ([16,18] and Appendix A). To isolate cell populations enriched in mouse mammary progenitor/stem cells and their derivative colony-forming cell progeny, we removed the hematopoietic and endothelial cell compartment and selected different mammary cell sub-compartments according to the expression of CD24 and CD29 cell surface markers. The CD24^+^/CD29^hi^ basal cell population is enriched in mammary stem/progenitor cells, whereas the CD24^+^/CD29^+^ luminal compartment contains luminal progenitor cells [31,33]. Figure 1A–C show that overexpression of PRB led to a statistically significant increase in the CD24^+^/CD29^hi^ basal and CD24^+^/CD29^+^ luminal cell compartments as compared to cells derived from mammary glands of wild type (WT) and PRA transgenic mice.

Next, we carried out mammosphere culture assays to establish whether there were changes in the proportions of cells with self-renewing capacity when PR isoform levels were modified. To do so, cells were cultured at low density in suspension cultures in serum-free, B27 and EGF supplemented media. A statistically significant increase in mammosphere-forming capacity was detected in the cell suspensions derived from the mammary glands of PRB transgenic mice; cells obtained from PRA transgenics did not show differences with those isolated from the glands of WT mice (Figure 1D,E). Together, these results suggest that an increase in the levels of PRB impacts the luminal and basal compartment and leads to an increase in cells with progenitor/stem cell characteristics.

### 3.2. Hormonal Regulation of the Luminal and Basal Compartments in Mammary Glands of WT and PR Transgenic Mice

To investigate the impact of hormone withdrawal on the stem/progenitor cell populations in the context of changes in the balance of PR isoforms, WT, PRA, and PRB mice were ovariectomized. Previously, Asselin-Labat et al. [6] showed that ovariectomy leads to a decrease in the luminal compartment. Flow cytometry analyses revealed that ovariectomy reduced the CD24^+^/CD29^+^ luminal cell compartment in mammary glands derived from WT and PRA mice; no changes were detected in cells derived from mammary glands of PRB mice (Figure 2A–C).

As expected, the basal population was stable upon ovariectomy, with no significant changes detected in the respective mammary glands as compared to the corresponding sham operated controls. Next, functional mammosphere assays were carried out. A statistically significant increase in mammosphere formation capacity was detected only in cell suspensions derived from PRA and WT mice upon OVX. This result is coherent with the decrease observed in the luminal compartment that would lead to a higher proportion of cells from the basal compartment enriched in stem/progenitor activity. Accordingly, no changes in mammosphere-forming capacity were observed in cell suspensions derived from PRB mice (Figure 2D,E).

To evaluate the impact of blocking estrogenic signaling, intact mice were treated with vehicle or ICI 182,780, a selective estrogen receptor downregulator [34]. We previously showed that in PRA transgenics, a four-day treatment with ICI 182,780 leads to a reversal of the hyperplastic phenotype as determined by the whole mount technique (Appendix A and [17]). Under these experimental conditions, control and treated mammary glands were dissociated and the resulting cell suspensions were analyzed by flow cytometry and mammosphere assays. Figure 3A–C show that blocking ER signaling did not have a significant impact on the basal or luminal compartments in mammary glands derived from WT or transgenic mice, as determined by flow cytometry. However, functional mammosphere assays revealed an increase in sphere-forming capacity in cells derived from glands of WT and PRA transgenic mice (Figure 3D,E).

No changes were observed in cells derived from PRB mice. This result suggests that in the case of WT and PRA mice, blocking estrogenic activity increases the proportion of stem/progenitor cells without affecting the overall size of the luminal and basal compartments.

Next, the direct impact of estrogen and ICI 182,780 was analyzed on mammosphere cultures derived from the mammary glands of untreated WT and transgenic mice. Under these conditions, an increase in sphere-forming capacity was only detected in ICI 182,780 treated cultures of cells derived from PRA mice (Figure 4A,B). No changes were detected in the cultures of cells derived from WT or PRB transgenics. On the other hand, estradiol had no impact on sphere cultures, independently of the genotypes.

Overall, these results suggest that blocking estrogen receptor signaling has a positive impact on the population of cells with self-renewing capacity, unless there is an increase in the levels of PRB.

### 3.3. Changes in PR Isoform Balance in Breast Cancer Cells Impacts on the Stem Cell Population

To study the impact of PR isoform balance on breast cancer stem cells, we used the previously characterized human T47D cell line that expresses both PR isoforms, and the sublines generated from it: T47D YA that exclusively expresses PRA and T47D YB that expresses PRB ([35] and Appendix A). Cells were characterized according to the percentage of CD44^+^/CD24^Low/−^ positive cells [32] and the mammosphere-forming capacity. Flow cytometry analysis revealed that expression of PRB led to a significant increase in the percentage of the CD44^+^/CD24^Low/−^ cells, in comparison to cells expressing PRA or both isoforms (Figure 5A,B). Next, tumorsphere assays were carried out. Alterations in PR isoform expression did not affect the sphere-forming capacity, as the three cell lines produced the same number of spheres in suspension cultures (Figure 5C,D), and this characteristic was maintained when secondary spheres were generated (not shown). Interestingly, even though the number of tumorspheres remained unchanged, T47D YB spheres were larger (*p* < 0.05) and irregular (*p* < 0.01) as compared to both T47D YA or the spheres derived from the parental cell line (Figure 5E,F).

These characteristics of tumorspheres have been previously related to a malignant phenotype [36]. Moreover, when cells from tumorspheres were challenged to a clonogenic assay, those derived from the YB cell line had a significantly higher clonogenic capacity than cells derived from the YA cell line, and similar to the parental one, further supporting the role of PRB in breast cancer malignancy (Figure 6).

Next, given that antiestrogens have been shown to increase the stem cell population in human breast cancer cells, we analyzed the impact of pre-treating cells for 72 h with vehicle, estradiol, or ICI 182,780 and subsequently, carried out flow cytometry and tumorsphere assays. Figure 7 shows that pre-treatment with estradiol significantly increased the percentage of CD44^+^/CD24^−/low^ cells only in the parental cell line. In mammosphere assays, we found that estradiol increased the number and size of tumorspheres in the parental and T47D YA cells, but did not impact T47D YB. The antiestrogen ICI 182,780 inhibited tumorsphere formation in number and size in the parental cell line only. Tumorspheres generated from T47D YB cells were not affected in number or size by these treatments (Figure 7C).

These results suggest that expression of the PRB isoform (in the absence of PRA) leads to independence of ER signaling. These observations are in line with different studies showing that PRA expressing tumors show a better response to tamoxifen treatment than those that overexpress PRB [26,27].

## 4. Discussion

In the current paper, we explored the impact of altering the PR isoform balance in the normal mouse mammary gland and in a human breast cancer cell line. Our results show that in the normal murine mammary gland, an increase in the levels of PRB leads to an increase in both luminal and basal cell compartments, together with an increase in cells with self-renewing capacity. Additionally, cells derived from PRB mice were not affected by ovariectomy or antiestrogen treatments either in vivo or in culture. In the T47D breast cancer cell line, expression of PRB increased the CD44^+^/CD24^Low/−^ subpopulation, and even though the number of tumorspheres did not vary between the T47D, YB, and YA cell lines, those generated from YB were bigger, more irregular and had increased clonogenic capacity. Additionally, the tumorsphere-forming capacity of T47DYB cells was not affected by pre-treatments with estrogens or antiestrogens. Together, our data suggest that changes in PR isoform balance regulate the stem cell compartment in the normal and neoplastic mammary gland, affecting its susceptibility to estrogenic regulation.

Previous reports exploring the participation of estrogen and progesterone on the stem cell population in human and murine normal breast have consistently pointed towards a key role for progesterone. Using a 3D culture model of human primary breast organoids, Graham et al. showed that treatment with progesterone led to an increase in progenitor cells as determined by mammosphere cultures and aldehyde dehydrogenase enzymatic activity [37]. Asselin-Labat and collaborators later on showed that in the normal mouse mammary gland, ovariectomy decreased the luminal CD24^+^CD29^+^ compartment but not the basal CD24^+^CD29^hi^ [6], as we report here. Additionally, they determined that pregnancy leads to a 24-fold increase in the basal CD24^+^CD29^hi^ cell population. Joshi et al. [7] showed that in diestrus, when progesterone levels are at their maximum, the pool of stem cells increases; similar results were found when mice were treated with estrogen plus progesterone after ovariectomy, but not with estrogen alone. Our results using WT mice are in the same line as these previous publications. However, when analyzing transgenic PRA and PRB mice, we found that there is an increase in the luminal and basal compartments in the mammary glands derived from PRB mice. Interestingly, studies analyzing the distribution of PRA vs. PRB in mouse mammary glands during development show that PRB is most highly expressed during pregnancy, with very few cells expressing both isoforms at this time [38,39]. In PRB knockout mice, there is a marked reduction in ductal branching and alveolar development during pregnancy, and RANKL induction is reduced [40]. Our results in the context of these previous publications suggest that PRB is critical in regulating stem cell expansion through the activation of RANKL. Other paracrine mediators of progesterone induced stem cell expansion have been proposed such as CXCL12/CXCR4 and WNT4/LRP5 [7,41]. However, there is no evidence as to whether they are regulated by PRB, although given the preponderance in the levels of expression of this particular isoform during stem/progenitor cell expansion, it could be a feasible possibility. Thus, it is plausible to hypothesize that the increase in stem and progenitor cells found in the mammary glands of PRB mice is related to the key role this isoform plays during stem cell expansion in pregnancy. Regarding the human mammary gland, there is limited information with regards to the fluctuations in PR isoform expression during development and pregnancy [38]. It has been established that both isoforms are co-expressed in the same cells and that the relative levels are similar during the menstrual cycle [24]. However, in depth studies of the changes in PR isoform expression during pregnancy are not available.

Our results analyzing the impact of PR isoforms using the T47D YA and YB cell lines, together with the parental T47D cells, show that expression of PRB on its own leads to an increase in CD44^+^/CD24^Low/−^ positive cells, larger tumorspheres, and increased clonogenic capacity. Tumorspheres generated from T47D YB cells were not affected in number or size by the treatments with estradiol, ICI 182,780, or tamoxifen, in comparison to those generated from T47D or T47D YA cells. These results are in discordance to those recently published by Truong et al. [42], where PRA is suggested as the driver of cancer stem cells. However, a close analysis at the methods used by their group shows that they differ in several key aspects to the ones used in this paper, thus, suggesting that they may not be discordant, but the result of different procedures. In particular, the cells were expanded in different media with different supplements; we used DMEM/F12 supplemented with 10% FBS and gentamicin, whereas Truong et al. used MEM containing 5% FBS, penicillin/streptomycin, MEM nonessential amino acids, and insulin [42]. The conditions in which the tumorspheres were generated were also different: as described here, we used B27 and EGF in phenol red-free DMEM/F12. In Truong’s paper, the culture media (phenol red-free DMEM/F12) were additionally supplemented with methylcellulose, insulin, hydrocortisone, and β-mercaptoethanol. Moreover, they determined the percentage of CD44^+^/CD24^Low/−^ cells from spheroids, whereas we set out from cells grown in monolayer, thus, not pre-selecting for those with self-renewal capacity, or their progeny. Thus, the behavior of the cells seems to be highly sensitive to the protocol used; cell biology experimental results have previously been shown to vary as a consequence of subtle changes in laboratory procedures [43]. Remarkably, a recent paper by Hosseini et al. shows that in a HER-2-driven mouse breast cancer model, progesterone triggers the migration of cancer cells from early lesions and that the B isoform of PR is associated with this process. Moreover, these early migrating cells displayed features of cancer stem cells [44,45]. Additionally, several reports point towards a key role for PRB in endocrine resistance: Sartorius et al. showed that xenografts generated from T47D YB cells did not respond to tamoxifen, whereas those from T47D YA cells grew slower and were responsive [46]. In samples derived from breast cancer patients, methylation of PRA, but not PRB, was predictive of lack of response to tamoxifen [47], and Wargon et al. showed in different in vivo cancer models that positive responsiveness to endocrine therapy was associated with overexpression of PRA [48]. Further studies are required to integrate our understanding of PR isoform signaling, cancer stem cells and response to treatment in breast cancer. This is especially relevant taking into account that PR isoforms have been shown to differentially reprogram ER signaling [49].

## 5. Conclusions

Our findings provide additional evidence for a role of progesterone signaling in regulating the homeostasis of normal and cancer stem cells. The role played by each PR isoform is still controversial, however, what does seem clear is that changes in the balance of PR isoforms has a direct impact on mammary gland biology and tumor progression.

## Figures and Tables

**Figure 1 cells-09-02074-f001:**
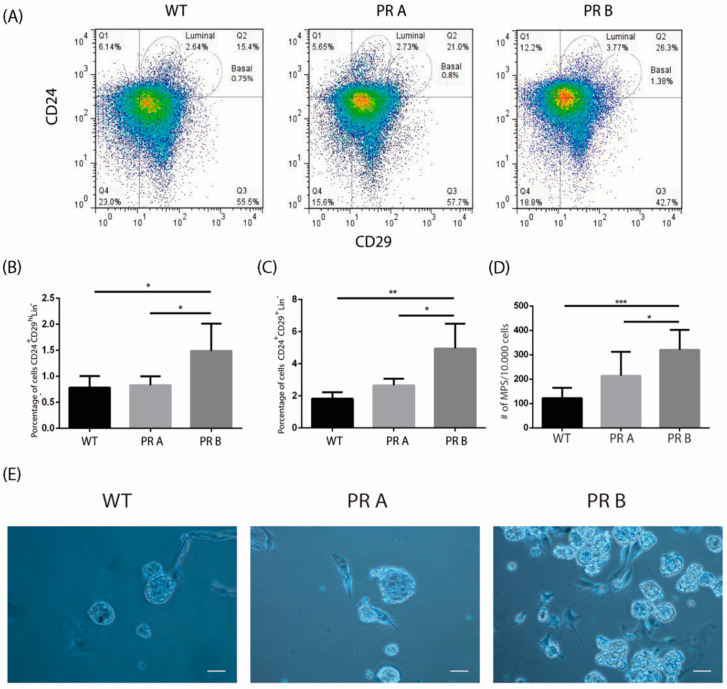
PRB overexpression alters the luminal and basal cell compartments in the normal mouse mammary gland. Mammary glands from intact WT, PRA, and PRB mice were collected, digested, and subjected to flow cytometry analysis or mammosphere suspension cultures. (**A**) Representative flow cytometries identifying CD24^+^CD29^low^ (luminal) and CD24^+^CD29^hi^ (basal) subpopulations. (**B**) Quantification of basal CD24^+^CD29^hi^ cells in mammary glands derived from WT, PRA, and PRB mice. The mean +/− SD of three independent experiments is shown. (**C**) Quantification of luminal CD24^+^CD29^low^ cells in mammary glands derived from WT, PRA, and PRBmice. The mean +/− SD of three independent experiments is shown. (**D**) Quantification of the number of mammospheres produced by cell suspensions derived from mammary glands of WT, PRA, and PRB mice. Once dissociated, cells were plated in serum-free medium supplemented with B27 and EGF in ultra-low attachment plates. Mammospheres were counted after 7–10 days in culture. The mean +/− SD of three independent experiments is shown. (**E**) Representative images of mammospheres generated from cells suspensions derived from WT, PRA, and PRB mice. Scale bar: 50 μm. *: *p* < 0.05; **: *p* < 0.01; ***: *p* < 0.001.

**Figure 2 cells-09-02074-f002:**
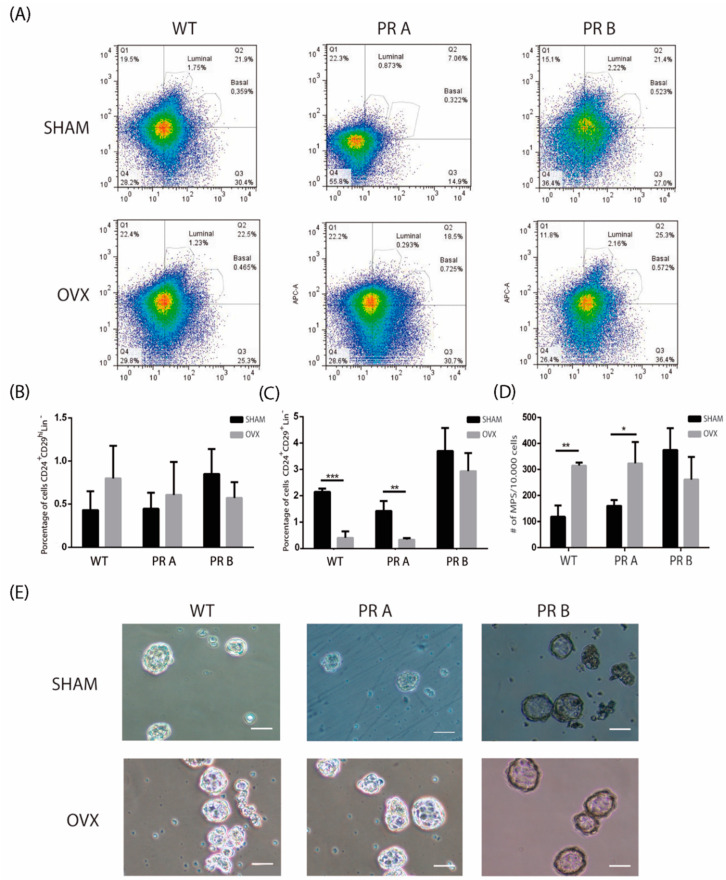
Impact of ovariectomy on the luminal and basal cell populations in mammary glands of wild type and PR transgenic mice. Mammary glands from intact or ovariectomized (OVX) mice were collected, digested, and subjected to flow cytometry analysis or mammosphere suspension cultures. (**A**) Representative flow cytometries identifying CD24^+^CD29^low^ (luminal progenitor) and CD24^+^CD29^hi^ (basal stem/progenitor) subpopulations. (**B**) Quantification of basal CD24^+^CD29^hi^ cells in mammary glands derived from sham operated and OVX WT, PRA, and PRB mice. The mean +/− SD of three independent experiments is shown. (**C**) Quantification of luminal CD24^+^CD29^low^ cells in mammary glands derived from sham operated and OVX WT, PRA, and PRB. The mean +/− SD of three independent experiments is shown. (**D**) Quantification of the number of mammospheres produced by cell suspensions derived from mammary glands of sham operated and OVX WT, PRA, and PRB mice. Mammospheres were counted after 7–10 days in culture. The mean +/− SD of three independent experiments is shown. (**E**) Representative images of mammospheres generated from cells suspensions derived from sham operated and OVX WT, PRA, and PRB mice. Scale bar: 50 μm. *: *p* < 0.05; **: *p* < 0.01; ***: *p* < 0.001.

**Figure 3 cells-09-02074-f003:**
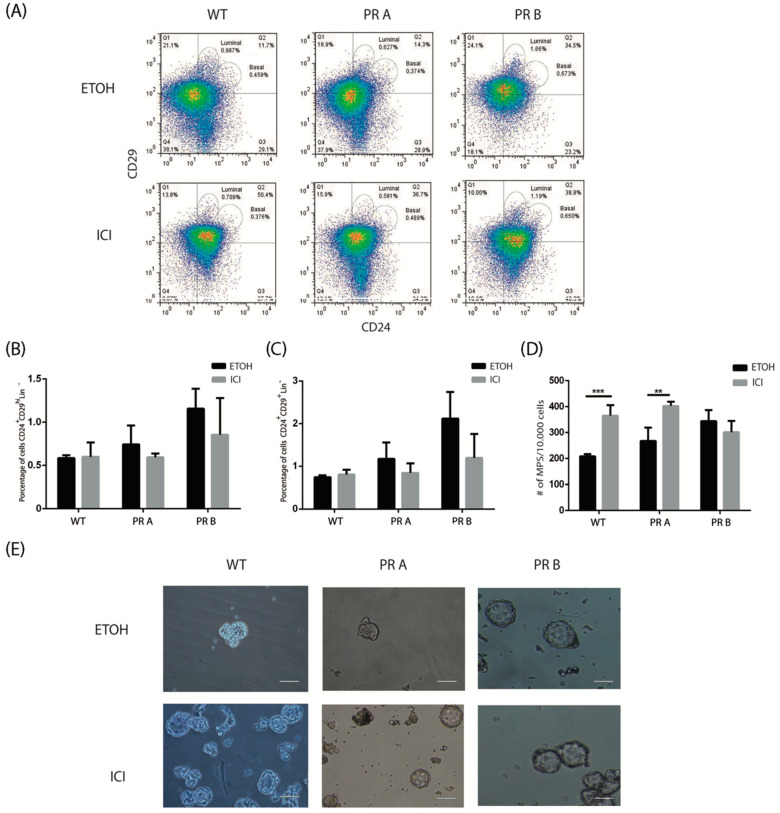
Impact of ICI 182,780 on the luminal and basal populations of mammary glands derived from WT and PR transgenic mice. Mammary glands derived from WT, PRA, and PRB mice treated with vehicle or ICI 182,780 (ICI) for four days were collected, digested, and subjected to flow cytometry or mammosphere suspension cultures. (**A**) Representative flow cytometries identifying CD24^+^CD29^low^ (luminal progenitor) and CD24^+^CD29^hi^ (basal stem/progenitor) subpopulations. (**B**) Quantification of basal CD24^+^CD29^hi^ cells in mammary glands derived from vehicle and ICI-treated WT, PRA, and PRB mice. The mean +/− SD of three independent experiments is shown. (**C**) Quantification of luminal CD24^+^CD29^low^ cells in mammary glands derived from vehicle and ICI-treated WT, PRA, and PRB. The mean +/− SD of three independent experiments is shown. (**D**) Quantification of the number of mammospheres produced by cell suspensions derived from mammary glands of vehicle and ICI-treated WT, PRA, and PRB mice. Mammospheres were counted after 7–10 days in culture. The mean +/− SD of three independent experiments is shown. (**E**) Representative images of mammospheres generated from cells suspensions derived from vehicle and ICI-treated WT, PRA, and PRB mice. Scale bar: 50 μm. **: *p* < 0.01; ***: *p* < 0.001.

**Figure 4 cells-09-02074-f004:**
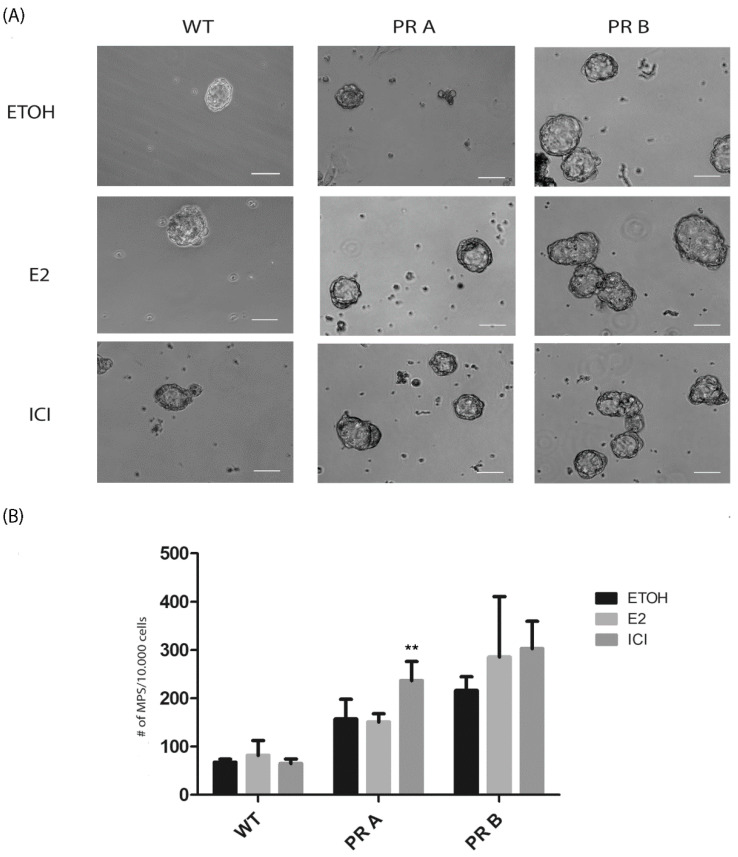
In vitro modulation of mammosphere-forming capacity by estradiol and ICI 182,780. Mammary glands from intact WT, PRA, and PRB mice were collected, digested, and plated under mammosphere culture conditions in the presence of vehicle, estradiol, or ICI 182,780 (ICI) as explained in the materials and methods. (**A**) Representative images of mammospheres derived from cell suspensions generated from mammary glands of WT, PRA, and PRB mice. (**B**) Quantification of the number of mammospheres produced by cell suspensions treated with vehicle, estradiol, or ICI. The mean +/− SD of three independent experiments is shown. Scale bar: 50 μm. **: *p* < 0.01.

**Figure 5 cells-09-02074-f005:**
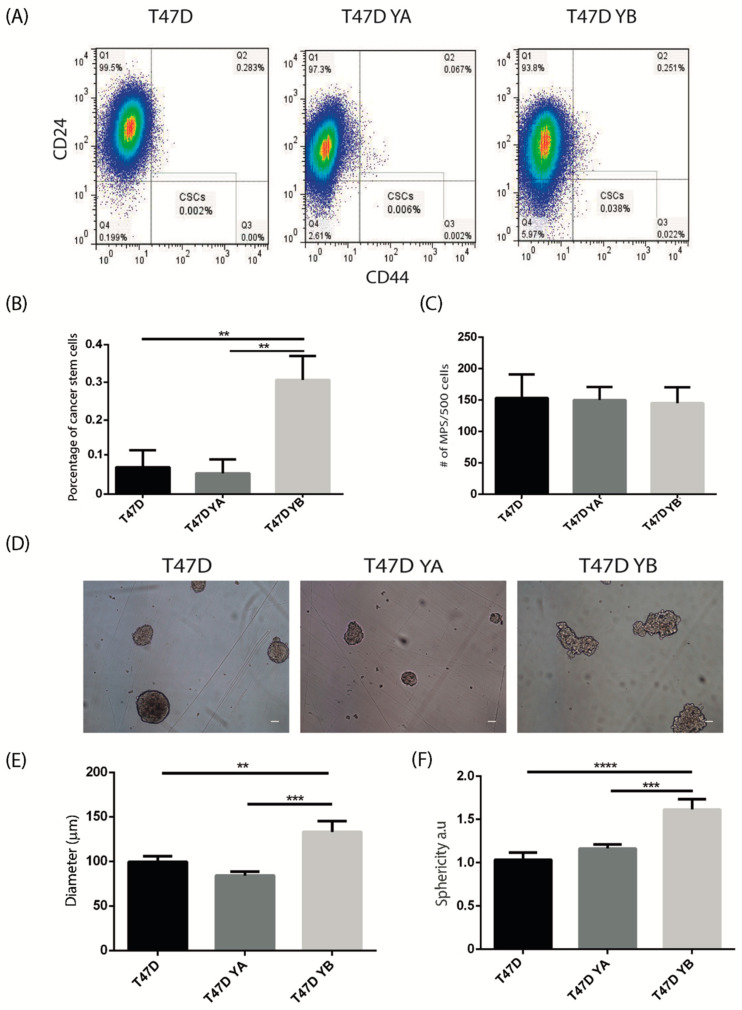
PRB is associated with an increase in the CD44^+^CD24^Low/−^ cell population in T47D human breast cancer cells. Parental T47D, YA, and YB cells were cultured as explained in the materials and methods, trypsinized and subjected to flow cytometry analysis for breast cancer stem cells markers CD44 and CD24. (**A**) Representative flow cytometries showing the CD44^+^CD24^Low/−^ stem cell population. (**B**) Mean +/− SD of the percentage of CD44^+^CD24^Low/−^ cells in parental T47D cells and YA and YB sublines. The data represent three independent experiments. (**C**) Quantification of the number of tumorspheres produced by cell suspensions derived from the parental T47D cells and the YA and YB sublines. Tumorspheres were generated by plating 500 cells/mL in serum-free medium, as explained in the materials and methods. They were counted after 7–10 days in culture. The mean +/− SD of three independent experiments is shown. (**D**) Representative images of tumorspheres generated by T47D, YA, and YB cell lines. (**E**) Quantification of the average diameter (in μm) of tumorspheres generated by the three cell lines. The mean +/− SD of three independent experiments is shown. (**F**) Average sphericity of tumorspheres generated by suspension cultures of T47D cells and YA and YB sublines was calculated as described in the materials and methods. The mean +/− SD of three independent experiments is shown. Scale bar: 50 μm. **: *p* < 0.01; ***: *p* < 0.001; ****: *p* < 0.0001.

**Figure 6 cells-09-02074-f006:**
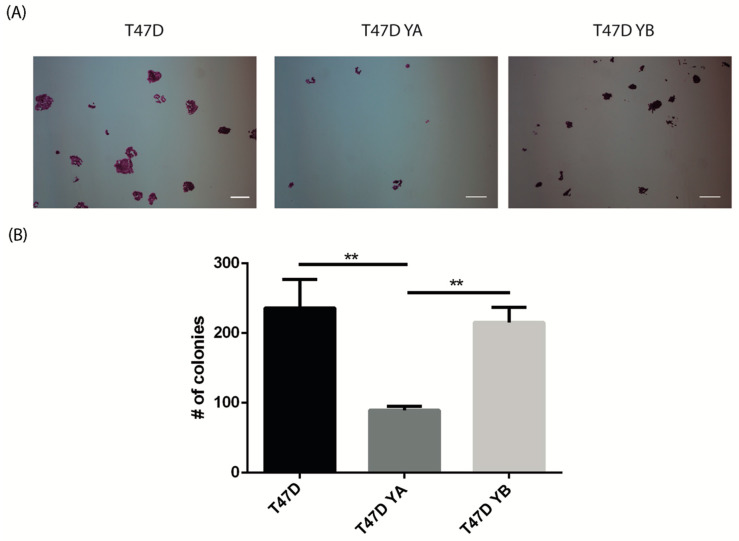
Cells derived from T47D YB tumorspheres have increased clonogenic capacity. Clonogenic assays were carried out by dissociating spheres derived from T47D, YA, and YB cell lines and plating the single cell suspensions at low density (10,000 cells/mL). (**A**) Representative images of clones obtained from T47D and T47D YA and YB cells. (**B**) Quantification of three independent clonogenic assays. The mean +/− SD of three independent experiments is shown. Scale bar: 50 μm. **: *p* < 0.01.

**Figure 7 cells-09-02074-f007:**
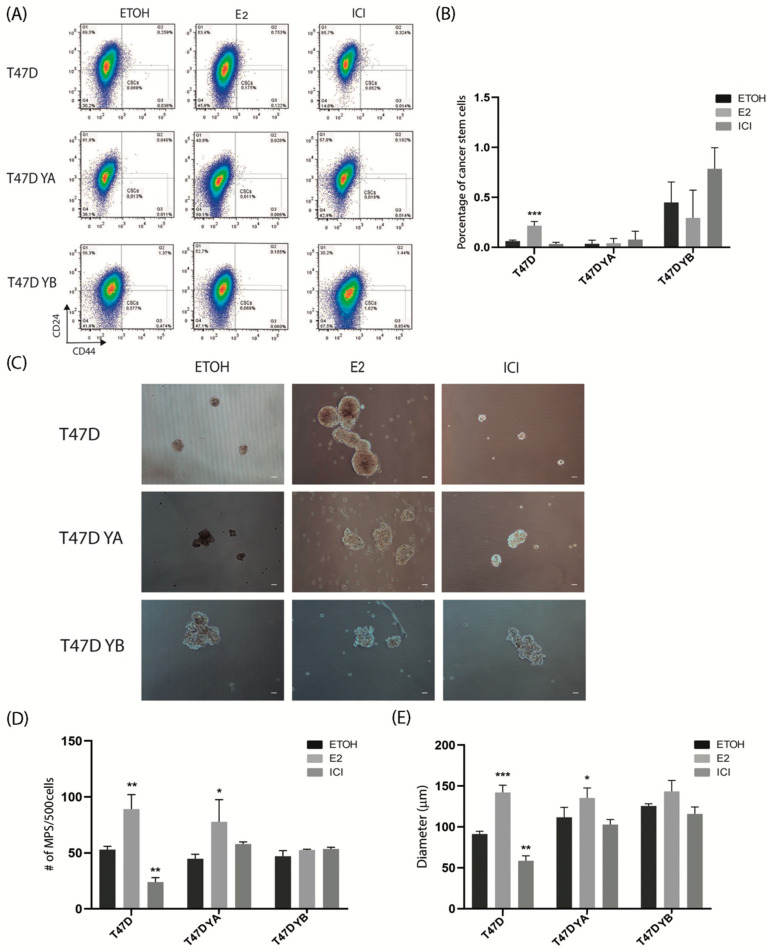
Impact of pre-treatment with estradiol and ICI 182,780 on breast cancer stem cells. Monolayers of T47D and T47D YA and YB cells were pre-treated with vehicle, estradiol (**E**) or ICI 182, 780 (ICI) for 72 h. They were then trypsinized and subjected to flow cytometry and tumorsphere cultures. (**A**) Representative flow cytometries of treated cells. (**B**) Quantification of three independent flow cytometries to identify CD44^+^CD24^Low/−^ cells. (**C**) Representative images of tumorspheres generated by T47D and T47D YA and YB cells after the corresponding pre-treatments. (**D**) Quantification of the number tumorspheres shown in C. (**E**) Quantification of the average diameter of tumorspheres shown in C. The mean +/− SD of three independent experiments is shown. Scale bar: 50 μm. *: *p* < 0.05; **: *p* < 0.01; ***: *p* < 0.001.

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
