# Peer review of "Alterations in Progesterone Receptor Isoform Balance in Normal and Neoplastic Breast Cells Modulates the Stem Cell Population"

_cells, 2020, doi:10.3390/cells9092074_

Round 1

Reviewer 1 Report

This paper by Recouvreux et al, presents data demonstrating that the progesterone isoform balance modulates the stem cell population in normal and neoplastic breast cell. The topic is important, and the authors have succeeded in forwarding the understanding of the development of breast cancer in a FVB mouse model together with the T47D cell culture model. The paper is extremely clearly written, and the results presented very nicely. However, there are a few minor points, which are listed below.

Minor points:

Abstract: The T47D derivatives YA and YB should be defined in the abstract.

P5 line 145 and 146: two spelling mistakes:

Nonspecific binding was addressed

Images were

P24 line 314: These characteristics… of tumorspheres …have been…

Author Response

We thank the reviewer for the constructive comments. The paper has been uploaded with the suggested corrections: 

1) Abstract: The T47D derivatives YA and YB should be defined in the abstract.

2) P5 line 145 and 146: two spelling mistakes:

Nonspecific binding was addressed

Images were

3) P24 line 314: These characteristics… of tumorspheres …have been…

Reviewer 2 Report

Manuscript ID: cells-906445 Title:Alterations in progesterone receptor isoform balance in normal and neoplastic breast cells modulates the stem cell population.   Recouvreux et al. investigated the role of PR isoforms on the homeostasis of stem cells in the normal mammary primary cells and cancer cells using PRA and PRB transgenic mice, and the breast cancer cell lines (T47D WT, YA and YB). They indicate that imbalance in PR isoform ratio has an impact on normal and tumorigenic breast progenitor/stem cells and suggest a key role for the B isoform with implications on response to anti-estrogens.   Their findings will be interested to the research community studying development of therapeutics for cancer prevention. 

This manuscript is well written to deliver their findings however the reviewer has several concerns for present version of the manuscript.

1. Figure 1. The mammosphere assay was performed in serum free media with EGF.  Why were FACS lineage-confirmed cells not used for mammosphere assay?  This will answer which lineage (luminal progenitors or stem cells) impacts the increase of cells in the context of PR isoform overexpression.

2. Authors studied effect of ER signal modulation on stem/progenitor expansion in the context of PR isoform imbalance in nulliparous mice. However they did not look into modulation of progesterone/PR signal in the current mouse model. Progesterone has direct effect on stem/progenitor expansion via RANKL/RANK and Wnt signaling due to its PR binding. At least authors need to explain why they did not explore addition of progesterone into experiment. If other researchers already explored this angle of stem cell expansion using PRA or PRB transgenic mice, please mention it in the discussion.  

3. In this mouse model, the authors demonstrated that PRB over-expressing cancer or normal cells are independent of estrogen/ER signaling regarding stem cell expansion whereas PRA overexpressing and WT cells are sensitive to estrogen/ER signaling. However PR modulates ER transcriptional activity, thus progesterone effect should be added for a full picture. It may be helpful to expand the discussion by adding current understanding of biology of ER/PR isoform interaction and influence to breast cancer risk.

4. Reviewer suggests that  the authors shall provide the audience some perspectives about whether blocking ER signaling in WT and PRA overexpressing cells are beneficial or make prognosis worse and how this information can be used for patient selection or time to be intervened.  

4. Minor corrections are required as following.

  • Methods of immunofluorescence and western blots were described however these ER or PR or E-cadherin images were not found in the manuscript. 
  • Supplementary figures are missing.
  • Please indicate when ovariectomy was performed in mice.

  • In each figure, there are several typos in p values. In the text, line 296 as well. Decimal point not comma should be used. 

  • In the text, a typo in line 125. 500.000 cells/ml should be corrected to 500,000 cells/ml.

  • Line 377-388: please rephrase the sentence to be clear that the majority of PR positive cells express only PRB "during pregnancy". PRA is the predominant isoform in nulliparous mammary glands in mice.

Author Response

We thank the reviewer for the constructive comments. Below are the responses and we have attached the paper with the suggested corrections. 

  1. Figure 1. The mammosphere assay was performed in serum free media with EGF.  Why were FACS lineage-confirmed cells not used for mammosphere assay?  This will answer which lineage (luminal progenitors or stem cells) impacts the increase of cells in the context of PR isoform overexpression.

We thank the reviewer for this suggestion, and we agree that FACS lineage-confirmed cells should have been used for mammosphere assays. These experiments were ongoing (before the pandemic) and they will be part of an upcoming manuscript. Still we believe that the data presented in this paper, showing an increase in cells with progenitor/stem characteristics in the mammary glands of PRB mice is relevant and an original finding.  We look forward to publishing this data in its current form, with future papers focusing on specific cell lineages.

  1. Authors studied effect of ER signal modulation on stem/progenitor expansion in the context of PR isoform imbalance in nulliparous mice. However they did not look into modulation of progesterone/PR signal in the current mouse model. Progesterone has direct effect on stem/progenitor expansion via RANKL/RANK and Wnt signaling due to its PR binding. At least authors need to explain why they did not explore addition of progesterone into experiment. If other researchers already explored this angle of stem cell expansion using PRA or PRB transgenic mice, please mention it in the discussion.  

We thank the reviewer for this interesting point. Progesterone, as mentioned, has been shown to modulate the expansion of stem/progenitor cells via RANKL/RANK and Wnt in the normal mammary gland by others (Asselin-Labat et al. 2010; Joshi et al., 2010; Graham et al 2009). We have not so far further explored the impact of treating the mice with progesterone in the transgenic setting because we were interested in focusing on how alterations in PR isoform balance impacted the stem cell population in general, and how this was modulated by anti-estrogens in view of the fact that ER is the main target for breast cancer treatment. However, we look forward to exploring how progesterone impacts the stem/progenitor cell population when PR isoform balance is modified in the normal murine mammary gland.

  1. In this mouse model, the authors demonstrated that PRB over-expressing cancer or normal cells are independent of estrogen/ER signaling regarding stem cell expansion whereas PRA overexpressing and WT cells are sensitive to estrogen/ER signaling. However, PR modulates ER transcriptional activity, thus progesterone effect should be added for a full picture. It may be helpful to expand the discussion by adding current understanding of biology of ER/PR isoform interaction and influence to breast cancer risk.

We thank the reviewer for this very interesting point. PR isoforms have been shown to differentially reprogram ER signaling (Singhal et al. 2018). Our results are in line with this notion and in particular with previous reports that show that when PRB is dominant, breast tumors do not respond to endocrine treatment both in experimental models and in patients. We have expanded the discussion to further make this point with the corresponding referenced. Please see in discussion, page 18, lines 428-436.

  1. Reviewer suggests that the authors shall provide the audience some perspectives about whether blocking ER signaling in WT and PRA overexpressing cells are beneficial or make prognosis worse and how this information can be used for patient selection or time to be intervened.  

We thank the reviewer for this additional comment. We believe that it is in line with the response to question 3. Our results support previous evidence that suggests that when PRB levels are increased or only PRB is expressed, response to endocrine therapy is lost and prognosis is worse.

Minor corrections:

1) Methods of immunofluorescence and western blots were described however these ER or PR or E-cadherin images were not found in the manuscript. 

We thank for the reviewer for this comment. The methods were in the context of the supplementary figures. We are sorry as there seems to have been a problem with the upload of the file. We have now included the supplementary figures in the same file.

2) Supplementary figures are missing.

We are sorry, this was the result of a technical problem.

3) Please indicate when ovariectomy was performed in mice.

We have added to the materials and methods the time of ovariectomy (20-22 weeks) (lines 89-91), page 4.  

4) In each figure, there are several typos in p values. In the text, line 296 as well. Decimal point not comma should be used. 

This has been corrected

5) In the text, a typo in line 125. 500.000 cells/ml should be corrected to 500,000 cells/ml.

Thank you, this has been corrected.

6) Line 377-388: please rephrase the sentence to be clear that the majority of PR positive cells express only PRB "during pregnancy". PRA is the predominant isoform in nulliparous mammary glands in mice.

We have corrected the sentence to make it clear that we are only referring to pregnancy (now line 390).